# Preliminary Design and Optimization of Axial Turbines Accounting for Diffuser Performance

**Roberto Agromayor * and Lars O. Nord**

Department of Energy and Process Engineering, NTNU—The Norwegian University of Science and Technology, Kolbj. Hejes v. 1B, NO-7491 Trondheim, Norway; lars.nord@ntnu.no
* Correspondence: roberto.agromayor@ntnu.no

**Abstract:** Axial turbines are the most common turbine configuration for electric power generation and propulsion systems due to their versatility in terms of power capacity and range of operating conditions. Mean-line models are essential for the preliminary design of axial turbines and, despite being covered to some extent in turbomachinery textbooks, only some scientific publications present a comprehensive formulation of the preliminary design problem. In this context, a mean-line model and optimization methodology for the preliminary design of axial turbines with any number of stages is proposed. The model is formulated to use arbitrary equations of state and empirical loss models and it accounts for the influence of the diffuser on turbine performance using a one-dimensional flow model. The mathematical problem was formulated as a constrained, optimization problem, and solved using gradient-based algorithms. In addition, the model was validated against two test cases from the literature and it was found that the deviation between experimental data and model prediction in terms of mass flow rate and power output was less than 1.2% for both cases and that the deviation of the total-to-static efficiency was within the uncertainty of the empirical loss models. Moreover, the optimization methodology was applied to a case study from the literature and a sensitivity analysis was performed to investigate the influence of several variables on turbine performance, concluding that: (1) the minimum hub-to-tip ratio constraint is always active at the outlet of the last rotor and that its value should be selected as a trade-off of aerodynamic performance and mechanical integrity; (2) that the total-to-static isentropic efficiency of turbines without diffuser deteriorates rapidly when the pressure ratio is increased; and (3) that there exist a loci of maximum efficiency in the specific speed and specific diameter plane (Baljé diagram) that can be predicted with a simple analytical expression.

**Keywords:** turbomachinery; mean-line; multistage; annular diffuser; gradient-based; loss model; baljé diagram; organic rankine cycle; cryogenic; supercritical carbon dioxide

## 1. Introduction

Axial turbines are the most common turbine configuration for electric power generation and propulsion systems, including: open Brayton cycles [1], closed Brayton cycles using helium [2] or carbon dioxide at supercritical conditions [3], and Rankine cycles using steam [4] or organic working fluids [5]. In addition, they are also used in cryogenic applications such as gas separation processes and liquefaction of natural gas [6]. Arguably, axial turbines owe their popularity to their versatility in terms of power capacity and range of operating conditions. The power capacity of axial turbines can vary from tens of kilowatts for small-scale Rankine power systems using organic fluids to hundreds of megawatts for large-scale steam and gas turbine power plants. In addition, the operating temperatures range from below $-200\,^\circ$C in some cryogenic applications to temperatures in excess of $1500\,^\circ$C for some advanced gas turbines, whereas the operating pressures can vary from a few millibars at the

exhaust of some steam turbines to hundreds of bars at the inlet of supercritical steam and carbon dioxide power systems.

The fluid-dynamic or aerodynamic design of axial turbines can be divided in several steps involving mathematical models of different levels of complexity ranging from low-fidelity models for the preliminary design (mean-line and through-flow models) to high-fidelity models for the detailed blade shape definition (solution of the Navier–Stokes equations with turbulence models) [7]. Even if mean-line models are the simplest approach to analyze the thermodynamics and fluid dynamics of turbomachinery, they are still an essential step of the fluid-dynamic design chain because they provide the information required to use more advanced flow models [8]. Mean-line models assume that the flow is uniform at a mean radius and evaluate the conditions at the inlet and outlet of each cascade using the balance equations for mass and rothalpy, a set of equations of state to compute thermodynamic and transport properties, and empirical loss models to evaluate the entropy generation within the turbine [9]. In addition, to automate the preliminary design, it is possible to formulate the mean-line model as an optimization problem. This is especially advantageous to design new turbine concepts because it allows exploration of the design space in a systematic way and account for technical limitations in the form of constraints [7].

Despite mean-line models being covered to some extent in turbomachinery textbooks [1,9], only some scientific publications present a comprehensive formulation of the preliminary design problem. Table 1 contains a non-exhaustive survey of mean-line axial turbine models in the open literature. Some of the differences in the model formulation include: considering single-stage or multistage turbines, using restrictive assumptions such as repeating stages or not, using simplified equations of state or real gas fluid properties, and whether or not to account for the influence of the diffuser on turbine performance. In addition, some works formulate the preliminary design as an optimization problem and then solve it using gradient-based or direct search optimization algorithms, while other works formulate the design problem as system of equations and then do parametric studies to explore the design space. One of the recurring limitations of the scientific literature is that most mean-line models have not been validated against experimental data or CFD simulations. Finally, to the knowledge of the authors, no publication has made the mean-line model source code openly available to the research community and industry, with the notable exception of the *Meangen* code by Denton [8].

In this work, a mean-line model and optimization methodology for the preliminary design of axial turbines with any number of stages is proposed. The model is presented in Section 2 and it was formulated to use arbitrary equations of state and empirical loss models and to account for the influence of the diffuser on turbine performance using a one-dimensional flow model from a previous publication [10]. In addition, Section 3 contains the validation of the model against experimental data from two well-documented test cases reported in the literature. After that, the design problem is formulated as a constrained optimization problem in Section 4 and the proposed optimization methodology was applied to a case study from the literature in Section 5 to assess the optimal design in terms of total-to-static efficiency, angular speed, and mean diameter. Finally, Section 6 contains a sensitivity analysis of the case study with respect to: (1) isentropic power output, (2) tip clearance height, (3) minimum hub-to-tip ratio, (4) diffuser area ratio, (5) diffuser skin friction coefficient, (6) total-to-static pressure ratio, (7) number of stages and (8) angular speed and mean diameter to gain insight about the impact of these variables on the performance of the turbine and to extract general design guidelines. The authors would like to mention that the source code with the implementation of the mean-line model and optimization methodology described in this paper is available in an online repository [11], see Supplementary Materials.

**Table 1.** Non-exhaustive survey of axial turbine mean-line models.

| Reference | Optimization [a] | Stages [b] | Repeating [c] | Diffuser [d] | Properties [e] | Validation [f] |
|---|---|---|---|---|---|---|
| Balje and Binsley [12] | Direct | 1 | No | No | Incompressible | Exp. |
| Macchi and Perdichizzi [13] | Direct | 1 | No | Fixed | Perfect gas | No |
| Lozza et al. [14] | Direct | 1, 2, 3 | No | Fixed | Perfect gas | No |
| Astolfi and Macchi [15] | Direct | 1, 2, 3 | No | Fixed | Real gas | No |
| Tournier and El-Genk [16] | No | Any | Yes | No | Real gas | Exp. |
| Da Lio et al. [17] | No | 1 | Yes | Fixed | Real gas | No |
| Da Lio et al. [18] | No | 1 | Yes | Fixed | Real gas | No |
| Meroni et al. [19] | Direct | 1 | No | Fixed | Real gas | Exp./CFD |
| Meroni et al. [20] | Direct | Any | No | Fixed | Real gas | Exp./CFD |
| Bahamonde et al. [21] | Direct | Any | No | No | Real Gas | CFD |
| Talluri and Lombardi [22] | Direct | 1 | Yes | No | Real gas | No |
| Denton [8] | No | Any | Optional | No | Perfect gas | CFD |
| *Present work* | Gradient | Any | Optional | 1D model | Real gas | Exp. |

[a] If applicable, type of optimization algorithm used (gradient-based or direct search). [b] Number of turbine stages that the model can handle. [c] Whether or not the model uses the repeating-stage assumption (this reduces the design space significantly). [d] Whether the model accounts for the influence of the diffuser or not. The models that accounted for the diffuser assumed a *fixed fraction* of kinetic energy recovery and did not model the flow within the diffuser. [e] Equation of state used to compute the properties of the fluid. [f] Whether or not the model has been validated with experimental data or CFD simulations.

## 2. Axial Turbine Model

Axial turbines are rotary machines that convert the energy from a fluid flow into work. An axial turbine is composed of one or more stages in series and each stage consists of one cascade of stator blades that accelerate the flow and one cascade of rotor blades that deflect the flow, converting the enthalpy of the fluid into work as a result of the net change of angular momentum. The kinetic energy of the flow at the outlet of the last stage can be significant and, for this reason, it is possible to install a diffuser to recover the kinetic energy and increase the turbine power output.

This section describes the axial turbine model proposed in this work. First, the geometry of axial turbines and the variables involved in the model are introduced. After that, the conventions used for the velocity triangles are explained. Finally, the design specifications (boundary conditions) and the mathematical model for the axial turbine is described. This mathematical model is composed of three sub-models that are used as building blocks: (1) the cascade model, (2) the loss model, and (3) the diffuser model. In Section 4 these sub-models are used to formulate the turbine preliminary design as a nonlinear, constrained optimization problem.

### 2.1. Axial Turbine Geometry

The geometry of a turbine blade is shown in Figure 1a. Blades are characterized by a mean camber line halfway between the suction and the pressure surfaces. The most forward point of the camber line is the leading edge and the most rearward point is the trailing edge. The blade chord $c$ is the length of the straight line connecting the leading and the trailing edges. The blade thickness is the distance between the pressure and suction surfaces, measured perpendicular to the camber line. The aerodynamic performance of a blade is influenced by the maximum thickness $t_{max}$ and the trailing edge thickness $t_{te}$. The angle between the axial direction and the tangent to the camber line is the metal angle $\theta$ and the difference between inlet and outlet metal angles is the camber angle $\Delta\theta = |\theta_{in} - \theta_{out}|$.

The axial–tangential view of a turbine stage is shown in Figure 1b. The blade pitch or spacing $s$ is the circumferential separation between two contiguous blades and the opening $o$ is defined as the distance between the trailing edge of one blade and the suction surface of the next one, measured perpendicular to the direction of the outlet metal angle. The angle between the axial direction and the chord line is the stagger angle or setting angle $\xi$ and the projection of the chord onto the axial direction is known as the axial chord $b$. The cascade spacing $s_c$ is the axial separation between one blade cascade and the next one.

The axial–radial view of a three-stage axial turbine is shown in Figure 1c. The working fluid flows parallel to the shaft within the annular duct defined by the inner and outer diameters. The hub is the surface defined by the inner diameter and the shroud is the surface defined by the outer diameter. The blade height $H$ is defined as the difference between the blade radius at the tip $r_t$ and the blade radius at the hub $r_h$ and the spacing between the tip of the rotor blades and the shroud is known as clearance gap height $t_{cl}$. The mean radius $r$ is often defined as the arithmetic mean of the hub and tip radii, although other definitions are possible. The blade height can vary along the turbine, but the flaring angle $\delta_{fl}$ should be limited to avoid flow separation close to the annulus walls. The geometry of an annular diffuser is shown in Figure 1d. The fluid leaving the last stage of the turbine enters the annular channel and it reduces its meridional component of velocity as the flow area increases (for subsonic flow) and its tangential component of velocity as the mean radius of the channel increases. The flow area of the diffuser is given by $A = 2\pi \hat{r} \hat{b}$, where $\hat{r}$ is the mean radius of the diffuser and $\hat{b}$ is the channel height of the diffuser. The area ratio is defined as the ratio of outlet to inlet areas, $AR = A_{out}/A_{in}$. When the inner and outer walls of the diffuser are straight, the diffuser is known as a conical-wall annular diffuser and its geometry can be parametrized in terms of the mean wall cant angle $\phi$ and the divergence semi-angle $\delta$.

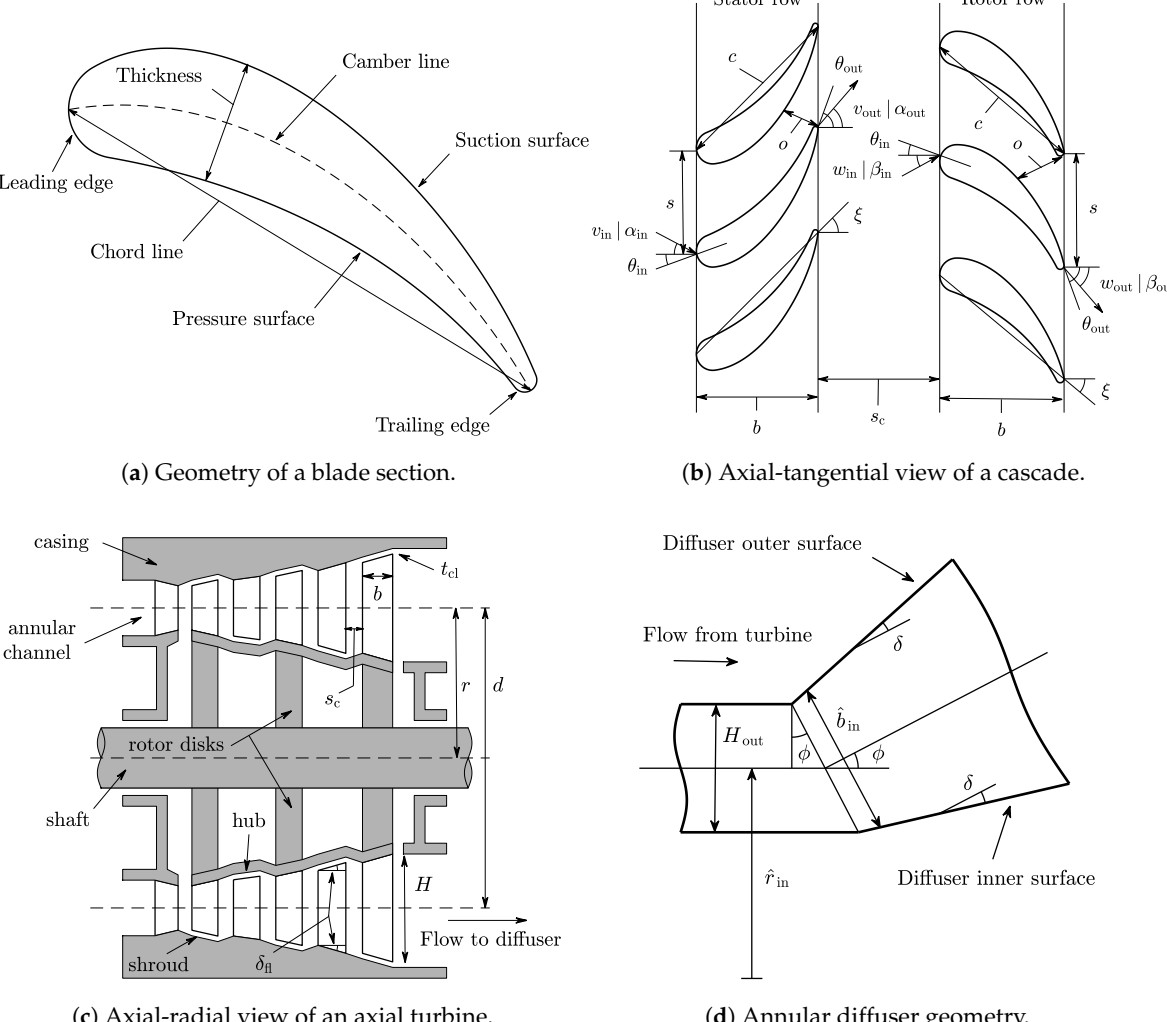

(**a**) Geometry of a blade section.

(**b**) Axial-tangential view of a cascade.

(**c**) Axial-radial view of an axial turbine.

(**d**) Annular diffuser geometry.

**Figure 1.** Geometry of a general axial turbine and exhaust diffuser.

## 2.2. Velocity Vector Conventions

In this work the symbol $v$ is used for the absolute velocity, $w$ for the relative velocity, and $u$ for the blade velocity. The components of velocity in the tangential and meridional directions are denoted with the subscripts $\theta$ and $m$, respectively. For the case of axial turbines, the meridional direction coincides with the axial direction. Regarding the sign convention for the velocity components, the positive axial direction is taken along the shaft axis from the inlet of the turbine to the outlet and the positive radial direction is taken as the turbine radius increases. The positive circumferential direction is taken in the direction of the blade speed.

The symbol $\alpha$ is used to denote the absolute flow angle while $\beta$ is used for the relative flow angle. As shown in the velocity triangle of Figure 2, all angles are measured from the meridional towards the tangential direction. This is the usual convention in the gas turbine industry and it bounds the flow angles to the interval $\left[-\frac{\pi}{2}, \frac{\pi}{2}\right]$ [1] (p. 316). The advantage of this angle convention is that it is possible to use single-input inverse trigonometric functions directly. In addition, the same sign convention is used for stator and rotor blades for the sake of consistency. However, the loss model that is used in this paper [23] employs a different sign convention for stator and rotor blades and some of the formulas of the loss model had to be adapted, see Appendix A.

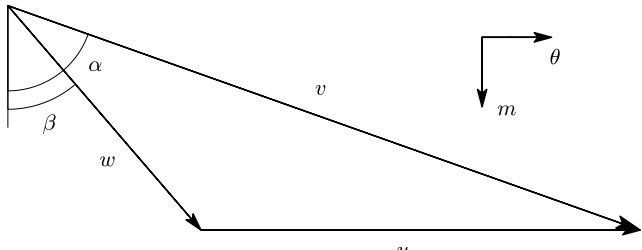

**Figure 2.** Velocity triangle showing the notation and conventions used in this work.

## 2.3. Design Specifications

A turbine is a component of a larger system that will impose some requirements on the turbine design, including: (1) stagnation state at the inlet of the turbine, (2) static pressure at the outlet of the turbine, and (3) mass flow rate. Alternatively, it is possible to specify the isentropic power output instead of the mass flow rate because both are related according to Equation (1), where the subscripts 1 and 2 refer to the states at the inlet and outlet of the turbine, respectively, and the subscript $s$ refers to an isentropic expansion.

$$\dot{W}_s = \dot{m}\,(h_{01} - h_{2s}) = \dot{m}\,\Delta h_s \tag{1}$$

These design requirements can be regarded as the thermodynamic boundary conditions for the expansion and they are given inputs for the turbine model.

## 2.4. Cascade Model

This section describes the equations used to model the flow within stator and rotor cascades. All flow variables are evaluated at constant mean radius at the inlet and outlet of each cascade (mean-line model). The cascade model presented in this section is solved sequentially and it contains three blocks: (1) computation of the velocity triangles, (2) determination of the thermodynamic properties using the principle of conservation of rothalpy and equations of state, and (3) calculation of the cascade geometry.

### 2.4.1. Velocity Triangles

The equations used to compute the velocity diagrams for rotor and stator sections are the same, provided that the blade velocity is given by $u = 0$ for the stators and $u = \omega\, r$ for the rotors, where the angular speed $\omega$ and mean radius $r$ are given as input variables.

The velocity triangles at the inlet of each cascade are computed according to Equations (2)–(7), where the subscripts that refer to inlet conditions have been dropped for simplicity. For the first stator, the absolute velocity $v$ and flow angle $\alpha$ are given as inputs. For the rest of cascades, the absolute velocity and flow angle are obtained from the outlet of the previous cascade.

$$v_\theta = v \sin(\alpha) \tag{2}$$

$$v_m = v \cos(\alpha) \tag{3}$$

$$w_\theta = v_\theta - u \tag{4}$$

$$w_m = v_m \tag{5}$$

$$w = \sqrt{w_\theta^2 + w_m^2} \tag{6}$$

$$\beta = \arctan\left(\frac{w_\theta}{w_m}\right) \tag{7}$$

The velocity triangles at the outlet of each cascade are computed according to Equations (8)–(13), where the subscripts that refer to outlet conditions have been dropped for simplicity. The relative velocity $w$ and flow angle $\beta$ at the outlet of each cascade are provided as an input for the model.

$$w_\theta = w \sin(\beta) \tag{8}$$

$$w_m = w \cos(\beta) \tag{9}$$

$$v_\theta = w_\theta + u \tag{10}$$

$$v_m = w_m \tag{11}$$

$$v = \sqrt{v_\theta^2 + v_m^2} \tag{12}$$

$$\alpha = \arctan\left(\frac{v_\theta}{v_m}\right) \tag{13}$$

### 2.4.2. Thermodynamic Properties

The axial turbine model was formulated in a general way and the thermodynamic properties of the working fluid can be computed with any set of equations of state that supports enthalpy–entropy function calls. In this work, the REFPROP fluid library was used for the computation of thermodynamic and transport properties [24].

The stagnation state at the inlet of the first stator (for instance temperature and pressure) is an input for the model and the corresponding static state is determined according to Equations (14) and (15). The remaining static properties at the inlet of the first stator are determined with enthalpy–entropy function calls to the REFPROP library, see Equation (16). The static thermodynamic properties at the inlet of all the other cascade are obtained from the outlet of the previous cascade.

$$h_{\text{in}} = h_{0,\text{in}} - \frac{1}{2}v_{\text{in}}^2 \tag{14}$$

$$s_{\text{in}} = s\left(p_{0,\text{in}},\, h_{0,\text{in}}\right) \tag{15}$$

$$[T,\, p,\, \rho,\, a,\, \mu]_{\text{in}} = [T,\, p,\, \rho,\, a,\, \mu]\left(h_{\text{in}},\, s_{\text{in}}\right) \tag{16}$$

The thermodynamic properties at the outlet of each cascade are computed using the fact that rothalpy is conserved both in rotor and stator cascades [9] (pp. 10–11). For purely axial turbines (constant mean radius), the conservation of rothalpy is reduced to the conservation of relative

stagnation enthalpy, and the static enthalpy at the outlet of each cascade can be computed according to Equation (17). In addition, the entropy at the outlet of each cascade is provided as an input to the model. Therefore, any other static thermodynamic property can be determined with enthalpy–entropy function calls to the REFPROP library, Equation (18).

$$h_{\text{out}} = h_{\text{in}} + \frac{1}{2}w_{\text{in}}^2 - \frac{1}{2}w_{\text{out}}^2 \tag{17}$$

$$[T, \, p, \, \rho, \, a, \, \mu]_{\text{out}} = [T, \, p, \, \rho, \, a, \, \mu]\,(h_{\text{out}}, s_{\text{out}}) \tag{18}$$

### 2.4.3. Cascade Geometry

The geometry of the annulus is obtained from the principle of conservation of mass and geometric relations, Equations (19)–(23), where the mass flow rate $\dot{m}$ is given as an input for the model. These equations are valid both for the inlet and outlet of the cascade and the subscripts were not included for simplicity.

$$A = \frac{\dot{m}}{\rho \, v_m} \tag{19}$$

$$H = \frac{A}{2\pi r} \tag{20}$$

$$r_{\text{h}} = r - H/2 \tag{21}$$

$$r_{\text{t}} = r + H/2 \tag{22}$$

$$\lambda = \frac{r_{\text{h}}}{r_{\text{t}}} \tag{23}$$

The mean blade height of the cascade is determined as the arithmetic mean of the blade height at the inlet and outlet of the cascade,

$$H = \frac{1}{2}\left(H_{\text{in}} + H_{\text{out}}\right) \tag{24}$$

The blade chord is determined from the blade aspect ratio (input variable) and the mean blade height using Equation (25). Similarly, the blade pitch (also known as spacing) is determined from the pitch to chord ratio (input variable) and the blade chord according to Equation (26).

$$c = \frac{H}{(H/c)} \tag{25}$$

$$s = c \cdot (s/c) \tag{26}$$

The incidence $i$ and deviation $\delta$ angles are assumed to be zero and, therefore, the metal angle at the inlet and outlet of each cascade are given by Equations (27) and (28), respectively.

$$\theta_{\text{in}} = \beta_{\text{in}} - i = \beta_{\text{in}} \tag{27}$$

$$\theta_{\text{out}} = \beta_{\text{out}} - \delta = \beta_{\text{out}} \tag{28}$$

The blade opening is given by Equation (29). This equation is an approximation that neglects the effect or the curvature of the blade suction surface [1] (pp. 343–344).

$$o \approx s \cdot \cos\left(\theta_{\text{out}}\right) \tag{29}$$

The maximum blade thickness is computed according to the formula proposed by Kacker and Okapuu [23] that correlates the blade maximum thickness to chord ratio with the camber angle as given by Equation (30). The blade camber angle is defined as $\Delta\theta = |\theta_{\text{in}} - \theta_{\text{out}}|$.

$$(t_{\text{max}}/c) = \begin{cases} 0.15 & \text{for} \quad \Delta\theta \leq 40° \\ 0.15 + 1.25 \cdot 10^{-3} \cdot (\theta - 40) & \text{for} \quad 40° \leq \Delta\theta \leq 120° \\ 0.25 & \text{for} \quad \Delta\theta \geq 120° \end{cases} \tag{30}$$

The stagger angle is computed according to Equation (31), which assumes that circular-arc blades are used, Dixon and Hall [9] (p. 72). Alternatively, the stagger angle could be computed using the graphical relation proposed by Kacker and Okapuu [23] or given as an input for the model. The axial chord of the blades is determined using the geometric relation given by Equation (32).

$$\xi = \frac{1}{2} (\theta_{\text{in}} + \theta_{\text{out}}) \tag{31}$$

$$b = c \cdot \cos(\xi) \tag{32}$$

The axial chord and blade height difference between inlet and outlet are used to compute the flaring angle of the cascade according to Equation (33).

$$\tan(\delta_{\text{fl}}) = \frac{H_{\text{out}} - H_{\text{in}}}{2\,b} \tag{33}$$

The trailing edge thickness is computed using the trailing edge thickness to opening ratio (input variable) and the cascade opening according to Equation (34).

$$t_{\text{te}} = o \cdot (t_{\text{te}}/o) \tag{34}$$

In addition, the axial spacing between cascades $s_{\text{c}}$ can be computed as fraction of the axial chord. However, this variable does not affect the performance predicted by the model because the Kacker and Okapuu [23] loss correlations neglect the influence of this parameter. Saravanamuttoo et al. [1] (pp. 332–333) suggests that axial spacing to chord ratios between 0.20 and 0.50 are satisfactory. Finally, the tip clearance height of the rotor blades $t_{\text{cl}}$ is given as a fixed input to the model that depends on manufacturing limits. The geometry relations presented in this section allow a description of the turbine geometry in a level of detail that is adequate for preliminary design purposes. A more detailed design of the turbine geometry, such as the definition of the shape of the blades, requires more advanced mathematical models based on the fluid dynamics within the turbine rather than the algebraic loss models used in this work.

### 2.5. Loss Model

During the preliminary design phase, it is common to use empirical correlations to estimate the losses within the turbine. These sets of empirical correlations are known as loss models. Losses can be interpreted as any mechanism that leads to entropy generation within the turbine (which in turn reduces the power output), such as viscous friction in boundary layers or shock waves. See the work by Denton [25] for a detailed description of loss mechanisms in turbomachinery.

Perhaps, the most popular loss model for axial turbines is the one proposed by Ainley and Mathieson [26,27] and its subsequent refinements by Dunham and Came [28] and Kacker and Okapuu [23]. The Kacker–Okapuu loss model has been further refined to account for off-design performance by Moustapha et al. [29] and by Benner et al. [30]. One of the remarkable aspects of the Ainley–Mathieson family of loss methods is that it has been updated with new experimental data several times since the first version of the method was published. This was not the case for

other loss prediction methods such as the ones proposed by Balje and Binsley [31], Craig and Cox [32], Traupel [33], or Aungier [34]. A comprehensive review of different loss models is given by Wei [35].

In this work, the Kacker and Okapuu [23] loss model was selected because of its popularity and maturity. The improvements of this loss model to account for off-design performance were not considered because the axial turbine methodology proposed in this paper is meant for the optimization of design performance. The Kacker–Okapuu loss model is described in detail in Appendix A. The formulation of the loss model has been adapted to the nomenclature and sign conventions used in this work for the convenience of the reader.

As described by Denton [25] and by Dahlquist [36], there are several definitions for the loss coefficient. In this work, the stagnation pressure loss coefficient was used because the Kacker–Okapuu loss model was developed using this definition. This loss coefficient is meaningful for cascades with a constant mean radius and it is defined as the ratio of relative stagnation pressure drop across the cascade to relative dynamic pressure at the outlet of the cascade, Equation (35). This definition is valid for both rotor and stator cascades.

$$Y = \frac{p_{0\text{rel,in}} - p_{0\text{rel,out}}}{p_{0\text{rel,out}} - p_{\text{out}}} \tag{35}$$

In general, the loss coefficient computed from its definition, Equation (35), and the loss coefficient computed using the loss model, Appendix A, will not match for an arbitrary set of input parameters. In Section 4, the turbine design is formulated as an optimization problem that uses equality constraints to ensure that the value of both loss coefficients matches for each cascade. The loss coefficient error is given by Equation (36).

$$Y_{\text{error}} = Y_{\text{definition}} - Y_{\text{loss model}} \tag{36}$$

*2.6. Diffuser Model*

The diffuser model is based on the transport equations for mass, meridional and tangential momentum, and energy in an annular channel. It assumes that the flow is one-dimensional (in the meridional direction), steady (no time variation), and axisymmetric (no tangential variation). The model can use arbitrary equations of state and it accounts for effects of area change, heat transfer, and friction. Under these conditions, the governing equations of the flow are given by Equations (37)–(40). The detailed derivation of these equations and a discussion of the physical meaning of the different terms is presented in the Appendix of Agromayor et al. [10].

$$v_m \frac{d\rho}{dm} + \rho \frac{dv_m}{dm} = -\frac{\rho v_m}{\hat{b}\,\hat{r}} \frac{d}{dm}(\hat{b}\,\hat{r}) \tag{37}$$

$$\rho v_m \frac{dv_m}{dm} + \frac{dp}{dm} = \frac{\rho v_\theta^2}{\hat{r}} \sin(\phi) - \frac{2\tau_w}{\hat{b}} \cos(\alpha) \tag{38}$$

$$\rho v_m \frac{dv_\theta}{dm} = -\frac{\rho v_\theta v_m}{\hat{r}} \sin(\phi) - \frac{2\tau_w}{\hat{b}} \sin(\alpha) \tag{39}$$

$$\rho v_m \frac{dp}{dm} - \rho v_m a^2 \frac{d\rho}{dm} = \frac{2(\tau_w v + \dot{q}_w)}{\hat{b} \left(\frac{\partial e}{\partial p}\right)_\rho} \tag{40}$$

The viscous term is modeled using a constant skin friction coefficient $C_f$ such that $\tau_w = C_f \frac{\rho v^2}{2}$ and heat transfer is neglected, $\dot{q}_w = 0$. The geometry of the diffuser was modeled in a simple way assuming that the inner and outer surfaces are straight, see Figure 1d. For this particular geometry, the diffuser channel height $\hat{b}$ and mean radius $\hat{r}$ are given by Equations (41) and (42), where the mean cant angle $\phi$ and divergence semi-angle $\delta$ are given as input parameters.

$$\hat{r}(m) = \hat{r}_{\text{in}} + m \sin(\phi) = r + m \sin(\phi) \tag{41}$$

$$\hat{b}(m) = \hat{b}_{\text{in}} + 2m \tan(\delta) = H_{\text{out}} / \cos(\phi) + 2m \tan(\delta) \tag{42}$$

The initial conditions required to integrate the system of ordinary differential equations (ODE) are prescribed assuming that the thermodynamic state and velocity vector do not change from the turbine outlet to the diffuser inlet. The integration starts from the initial conditions and stops when the prescribed value of outlet to inlet area ratio $AR$ is reached. In this work, the MATLAB function *ode45* was used to perform the numerical integration [37]. This function uses an automatic-stepsize-control solver that combines fourth and fifth order Runge–Kutta methods to control the error of the solution.

In general, the static pressure that is given as a design specification from a system analysis will not match the pressure at the outlet of the diffuser computed by the model. In Section 4, the turbine design is formulated as an optimization problem that uses an equality constrain to ensure that the static pressure at the outlet of the diffuser and the target pressure match. The dimensionless outlet static pressure error is given by Equation (43).

$$p_{\text{error}} = \frac{p_{\text{out}}^{\text{diff}} - p_{\text{target}}}{p_{\text{target}}} \tag{43}$$

## 3. Validation of the Axial Turbine Model

The aim of this section is to validate the axial turbine model presented in Section 2 using the experimental data of the one- and two-stage turbines reported by Kofskey and Nusbaum [38]. The flow in both turbines is subsonic and they use air as working fluid. To validate the model, the geometry and operating conditions reported by Kofskey and Nusbaum [38] were replicated and the design-point performance of both test cases was compared with the output of the model, see Table 2. The inlet thermodynamic state, angular speed, and total-to-static pressure ratio were matched at the design point and the validation was performed analyzing the deviation in mass flow rate, power output, and total-to-static isentropic efficiency. This approach is consistent with the definition of the design point given by Kofskey and Nusbaum [38]. The data reported in Table 2 shows that the agreement between the predicted and measured mass flow rates and power outputs is satisfactory and that the relative deviation is less than 1.2% for both turbines. In addition, the deviation of total-to-static isentropic efficiency between model and experiment is 1.15 percentage points for the one-stage turbine and 0.60 points for the two-stage turbine, which is within the efficiency-prediction uncertainty of the loss model of $\pm 1.5$ percentage points [23]. The turbines reported in [38] did not have a diffuser to recover the exhaust kinetic energy and therefore could not be used to validate the diffuser model. Nevertheless, the diffuser model used in this work has been validated in a previous publication [10].

The analysis presented in this section showed that the axial turbine model can be used to predict the design-point performance of turbines with one or more stages. However, the validation was restricted to subsonic turbines using air as working fluid and it is likely that the efficiency predictions will not be as accurate for turbines with transonic–supersonic cascades or when the fluid behavior deviates from ideal gas, such as in Rankine cycles using organic fluids with high molecular mass or in supercritical carbon dioxide power systems.

**Table 2.** Validation of the axial turbine model against experimental data.

| Number of Stages | Variable [a,b,c] | Kofskey and Nusbaum [38] | Present Work | Deviation |
|---|---|---|---|---|
| 1 stage | $T_{01}$ | 22.5 °C | Same | n.a. |
| | $p_{01}$ | 1.380 bar | Same | n.a. |
| | $PR$ | 2.298 | Same | n.a. |
| | $\omega$ | 15,533 rpm | Same | n.a. |
| | $\dot{m}$ | 2.695 kg/s | 2.729 kg/s | 0.91% |
| | $\dot{W}$ | 136.17 kW | 135.03 kW | −0.42% |
| | $\eta_{ts}$ | 80.00% | 78.85% | 1.15 points |
| 2 stages | $T_{01}$ | 25.8 °C | Same | n.a. |
| | $p_{01}$ | 1.240 bar | Same | n.a. |
| | $PR$ | 4.640 | Same | n.a. |
| | $\omega$ | 15,619 rpm | Same | n.a. |
| | $\dot{m}$ | 2.407 kg/s | 2.434 kg/s | 1.12% |
| | $\dot{W}$ | 212.06 kW | 211.10 kW | −0.46% |
| | $\eta_{ts}$ | 82.00% | 81.40% | 0.60 points |

[a] Kofskey and Nusbaum [38] reported the turbine performance in terms of *equivalent* variables. These were converted to ordinary variables using ambient conditions at sea level (101.325 kPa and 288.15 K). [b] The power output is computed from the measured torque and angular speed. [c] The total-to-static isentropic efficiency is a dependent variable that is computed from the thermodynamic conditions, mass flow rate, and power output.

## 4. Optimization Problem Formulation

The sub-models presented in Section 2 can be integrated to formulate the design of the turbine as a nonlinear, constrained optimization problem. To formulate this problem, it is necessary to specify: (1) the objective function to be optimized, (2) the independent variables and fixed parameters, and (3) the inequality and equality constraints that limit the design space. Once the problem is formulated, it is possible to find the optimal solution that satisfies the constraints using a numerical algorithm.

### 4.1. Objective Function

The objective function is any indicator of interest that must be minimized or maximized. In this work the total-to-static isentropic efficiency was used as objective function because it is assumed that the kinetic energy at the outlet of the diffuser is wasted [9] (pp. 23–24). The total-to-static isentropic efficiency is given by Equation (44), where the subscripts 1 and 2 refer to the states at the inlet and outlet of the turbine, respectively, and the subscript *s* refers to an isentropic expansion.

$$\eta_{ts} = \frac{h_{01} - h_{02}}{h_{01} - h_{2s}} \tag{44}$$

### 4.2. Independent Variables

The choice of independent variables is not unique and different sets of variables can be used to formulate the same problem. Ideally, it should be easy to provide a set of independent variables and this set should allow computation of the dependent variables in a sequential way, avoiding iterations when the model is evaluated. In addition, it is preferable to use a set of independent variables that is not poorly scaled, Nocedal and Wright [39] (pp. 26–27). An appropriate scaling of the problem increases the convergence rate of some algorithms and will reduce the numerical rounding error when the gradient of the objective function and constraints are evaluated using finite differences.

Table 3 shows the set of independent variables adopted in this work as well as the lower and upper bounds that were used to formulate the optimization problem for the case study discussed in Section 5. This formulation uses 6 independent variables per cascade (or 12 independent variables per stage) plus three additional global independent variables. This set of independent variables is well-scaled (all variables have a similar order of magnitude) and it allows evaluation of the turbine model in a sequential way without inner iterations from the inlet of the first stator to the exit of the

diffuser. The lower and upper bounds of the independent variables were selected to span a large design space that respects the range of applicability of the Kacker and Okapuu [23] loss system.

The specific speed and specific diameter were selected as independent variables instead of the angular speed and mean diameter because they have an order of magnitude of unity and it is easier to provide a reasonable initial guess since they are independent of the scale of the problem [31]. The definitions of the specific speed and diameter are given by Equations (45) and (46). The angular speed and mean diameter can be readily obtained from their specific counterparts.

**Table 3.** Optimization problem formulation and definition of the reference cases.

| Fixed parameters | | | | |
|---|---|---|---|---|
| Number of stages | $N$ | $=$ | 1 | – |
| Isentropic power output | $\dot{W}_s$ | $=$ | 250\|5000 | kW |
| Turbine inlet stagnation temperature | $T_{01}$ | $=$ | 155 | °C |
| Turbine inlet stagnation pressure | $p_{01}$ | $=$ | 36.2 | bar |
| Turbine outlet static pressure | $p_2$ | $=$ | 15.85 | bar |
| Tip clearance height | $t_{cl}$ | $=$ | 0.50 | mm |
| First stator inlet flow angle | $\alpha_{in}$ | $=$ | 0.0 | deg |
| Diffuser mean cant angle | $\phi$ | $=$ | 30.0 | deg |
| Diffuser divergence semi-angle | $\delta$ | $=$ | 5.0 | deg |
| Diffuser area ratio | $AR$ | $=$ | 2.5 | – |
| Diffuser skin friction coefficient | $C_f$ | $=$ | 0.010 | – |
| Independent variables | | | | |
| Specific speed | $\omega_s$ | $\in$ | [0.10, 10.0] | – |
| Specific diameter | $d_s$ | $\in$ | [0.10, 10.0] | – |
| Normalized first stator inlet velocity | $v_{in}/v_0$ | $\in$ | [0.01, 1.00] | – |
| Normalized outlet relative velocity [a] | $w_{out}/v_0$ | $\in$ | [0.01, 1.00] | – |
| Outlet relative flow angle (stator) [a] | $\beta_{out,S}$ | $\in$ | [+40.0, +80.0] | deg |
| Outlet relative flow angle (rotor) [a] | $\beta_{out,R}$ | $\in$ | [−80.0, −40.0] | deg |
| Normalized outlet entropy [a,c] | $s_{out}/s_{in}$ | $\in$ | [1.00, $s_{ref}/s_{in}$] | – |
| Aspect ratio [a] | $H/c$ | $\in$ | [1.00, 2.00] | – |
| Pitch to chord ratio [a] | $s/c$ | $\in$ | [0.75, 1.10] | – |
| Trailing edge thickness to opening ratio [a] | $t_{te}/o$ | $\in$ | [0.05, 0.40] | – |
| Nonlinear constraints | | | | |
| Inlet relative flow angle (stator) [a] | $\beta_{in,S}$ | $\leq$ | +15.0 | deg |
| Inlet relative flow angle (rotor) [a] | $\beta_{in,R}$ | $\geq$ | −15.0 | deg |
| Flaring angle [a] | $\delta_{fl}$ | $\in$ | [−10.0, +10.0] | deg |
| Hub-to-tip ratio [b] | $\lambda$ | $\in$ | [0.60, 0.95] | – |
| Cascade pressure ratio [a] | $PR_c$ | $\geq$ | 1.00 | – |
| Diffuser inlet meridional Mach number | $Ma_{m,in}^{diff}$ | $\leq$ | 1.00 | – |
| Outlet static pressure error | $p_{error}$ | $=$ | 0.00 | – |
| Cascade loss coefficient error [a] | $Y_{error}$ | $=$ | 0.00 | – |

[a] One value per cascade (rotor or stator). [b] Two values per cascade (inlet and outlet). [c] $s_{ref}$ corresponds the outlet entropy assuming $\eta_{ts}^{ref} = 50\%$.

$$\omega_s = \omega \frac{(\dot{m}/\rho_{2s})^{1/2}}{(h_{01} - h_{2s})^{3/4}} \tag{45}$$

$$d_s = d \frac{(h_{01} - h_{2s})^{1/4}}{(\dot{m}/\rho_{2s})^{1/2}} \tag{46}$$

In addition, the flow velocities were normalized using the isentropic velocity (also known as spouting velocity), see Equation (47), and the entropies were normalized using a reference entropy value computed assuming an isentropic turbine efficiency, see Equations (48) and (49).

$$v_0 = \sqrt{2\,(h_{01} - h_{2s})} \tag{47}$$

$$s_{\text{ref}} = s(p_2, h_{\text{ref}}) \tag{48}$$

$$h_{\text{ref}} = h_{01} - \eta_{\text{ref}} \cdot (h_{01} - h_{2s}) \tag{49}$$

The remaining independent variables did not need to be scaled.

### 4.3. Fixed Parameters

To compute the dependent variables, it is also necessary to provide several fixed parameters that will not change during the optimization. Table 3 contains the set of fixed parameters that are used as input for the axial turbine model as well as the numerical values used for the case study.

The isentropic power output, given by Equation (1), and thermodynamic boundary conditions are kept constant because they are given by the system design requirements, see Section 2.3. It is interesting to note that the set of independent variables is dimensionless and that the isentropic power output (or the mass flow rate) is the variable that scales-up the problem.

The flow angle at the inlet of the first stator was not selected as an optimization variable because it was assumed that there is no swirl at the inlet of the turbine and the tip clearance height was set as a fixed parameter because it is given by manufacturing limits. Finally, the diffuser model inputs were set as fixed parameters because the total-to-static isentropic efficiency is a monotonic function of these variables, see Agromayor et al. [10].

### 4.4. Constraints

In addition to the bounds for the degrees of freedom, the axial turbine model includes nonlinear equality and inequality constraints to guarantee that the model is consistent and that the design is feasible. These constraints and the numerical values used for the case study are summarized in Table 3. Depending on each application, it is possible to implement additional constraints for the model or to ignore some of the constraints suggested in Table 3. The pressure ratio in each cascade was constrained to avoid compression within the turbine (this is equivalent to constrain the degree of reaction between zero and one) and the constraint on the flaring angle was set to avoid flow separation close to the annulus walls [26]. In addition, the constrains for the inlet relative flow angles were imposed to avoid blades with too low deflection [23]. The meridional component of the Mach number at the inlet of the diffuser was constrained to ensure that the ODE system of the diffuser model, Equations (37)–(40), is not singular [10].

The constrain for the hub-to-tip ratio was set because the hub-to-tip ratio is a dependent variable that the designer might want to control since it has a great influence on the optimal angular speed and diameter, on the aerodynamic design of the blades (radial variation of flow angles), and on the centrifugal and gas bending stresses [1].

Finally, as discussed in Sections 2.5 and 2.6, equality constrains were imposed on the loss coefficient error and the outlet static pressure error to ensure that the model is consistent.

### 4.5. Optimization Algorithm

The are two main families of optimization methods, gradient-based method and direct search or derivative-free methods. Gradient-based methods are suitable for problems that are continuous and smooth, and they use derivative information of the objective function and constraints to determine the next iterate, while direct search methods use only sampled values of the objective function and constraints. For this reason, direct search methods can be used when gradient information

is not available or to solve non-smooth or even discontinuous problems. However, direct search methods have slower convergence rates and they are not well suited to handle problems with equality constraints [39] (pp. 220–223).

The axial turbine model proposed in this work is continuous and smooth, except for some of the equations of the loss model that are not differentiable (piece-wise functions or functions involving absolute values), see Appendix A. Despite these non-smooth points, experience shows that the optimization problem can be solved successfully using gradient-based optimization algorithms. In particular, the sequential quadratic programming algorithm of the MATLAB Optimization Toolbox was used to optimize the axial turbine design [40].

*4.6. Optimization Strategy*

Once the optimization problem is formulated it is possible to find the optimal solution that satisfies the constraints using a numerical algorithm. The optimization strategy for axial turbines is summarized in Figure 3. The optimization requires an initial guess for the independent variables and the values of the fixed parameters listed in Table 3. In addition, it is necessary to specify the choice of optimization algorithm and the tolerances for the termination criteria. In the first iteration, the optimization algorithm uses the initial guess to evaluate the axial turbine model. Once the axial turbine model is evaluated, the gradients of the objective function and the constraints are calculated by finite differences and they are used by the optimization algorithm to determine the values of the independent variables in the next iteration. This process is repeated until the solution meets the termination criteria and the optimal turbine design is saved.

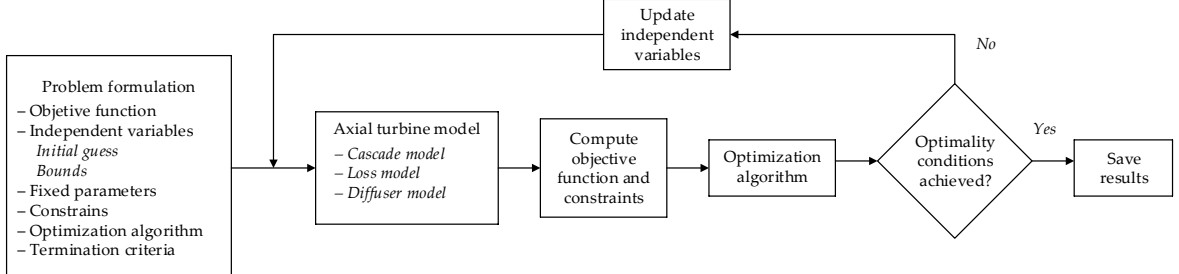

**Figure 3.** Axial turbine optimization strategy flow diagram.

## 5. Optimization of a Case Study

The aim of this section is to assess the optimization methodology proposed in this work. To do this, the model was tested against two reference axial turbine optimization problems presented in Macchi and Astolfi [41]. The two cases consider pentaflueroethane (R125) expanding from 155 °C and 36.2 bar (stagnation properties) to 15.85 bar (static pressure). The mass flow rate is selected to achieve an isentropic power of 250 kW in the first case and 5000 kW in the second case. These two case studies are representative of a small-scale and a large-scale Rankine cycles used to generate power from a low-temperature heat source as in a geothermal, solar, biomass, or waste heat recovery application [5].

The values of the fixed parameters, bounds of the optimization variables, and the nonlinear constrains used to formulate the optimization problem are summarized in Table 3. The minimum hub-to-tip ratio constraint is always active at the exit of the last rotor, see Section 6.3, and it has a great influence on the optimal angular speed and diameter. For this reason, the comparison of optimal speed and diameter will only be fair if the minimum hub-to-tip ratio is the same as in the reference case. The value $\lambda_{\min} = 0.60$ reported in Table 3 is the same value used by Macchi and Astolfi [41] (the minimum-hub-to tip ratio used by Macchi and Astolfi was not reported in the original publication, but it was confirmed by M. Astolfi in a personal communication).

The results of the optimization for the two cases considered are shown in Table 4. It can be observed that the optimal angular speed and diameter of the reference case agree well with the results obtained with the turbine model presented in this work and that the maximum relative deviation is less than 6%. In addition, the model presented in this work captures the trend of the angular speed and diameter as the power output changes. The values of total-to-static efficiency from the reference case and the ones obtained in the present work are comparable although there are is a difference of 2.07 and 0.83 percentage points in the small-scale and large-scale cases, respectively. This difference is not surprising considering that the Craig and Cox [32] loss model was used in the reference case while the Kacker and Okapuu [23] model was used in the present work and that the efficiency-prediction uncertainty of these empirical loss models is approximately ±1.5 percentage points, if not higher [23].

**Table 4.** Output of the optimization methodology for two reference optimization problems.

| Isentropic Power | Variable | Macchi and Astolfi [41] | Present Work | Deviation |
|---|---|---|---|---|
| $\dot{W}_s = 250$ kW | $\omega$ | 31,000 rpm | 29,231 rpm | −5.71% |
| | $d$ | 0.086 m | 0.087 m | 1.27% |
| | $\dot{W}$ | 219.3 kW | 224.4 kW | 2.36% |
| | $\eta_{ts}$ | 87.70% | 89.77% | 2.07 points |
| $\dot{W}_s = 5000$ kW | $\omega$ | 6000 rpm | 6144 rpm | 2.40% |
| | $d$ | 0.420 m | 0.395 m | −5.87% |
| | $\dot{W}$ | 4535.0 kW | 4576.5 kW | 0.92% |
| | $\eta_{ts}$ | 90.70% | 91.53% | 0.83 points |

## 6. Sensitivity Analysis

This section contains a sensitivity analysis of the 5000 kW reference case analyzed in the previous section to gain insight about the impact of several input parameters on turbine performance. The next subsections investigate the influence of: (1) isentropic power output, (2) tip clearance height, (3) minimum hub-to-tip ratio, (4) diffuser area ratio, (5) diffuser skin friction coefficient, (6) total-to-static pressure ratio, (7) number of stages and (8) angular speed and mean diameter on the total-to-static isentropic efficiency. Each of the analyses of this section studies the influence of these variables on the optimal solution while all other fixed parameters are the same as in the 5000 kW reference case summarized in Table 3. The ranges of the variables were selected to cover a wide span of flow conditions and they are justified in each subsection. Other variables were not considered for the sensitivity analysis because they have a secondary influence on turbine performance or because they are inactive constraints.

### 6.1. Influence of Isentropic Power

The isentropic power output was varied from 10 kW to 10 MW and the maximum attainable total-to-static efficiency is shown in Figure 4a,b. This range of power output was selected to cover a wide spectrum of turbine scales. According to the classification for Rankine power systems using organics working fluids proposed by Colonna et al. [5], this range of power output covers the mini, small, medium, and large power capacities.

It can be observed that the efficiency increases monotonously with the isentropic power and that the effect is more marked when the power output is small. The reason for this is that the size of the turbine increases and therefore: (1) the blade height $H$ increases and the $t_{cl}/H$ ratio decreases, which in turn reduces the tip clearance loss coefficient, see Equation (A19), and (2) the blade chord of the cascades increases, which in turn increases the Reynolds number and reduces the profile loss coefficient, see Equation (A1).

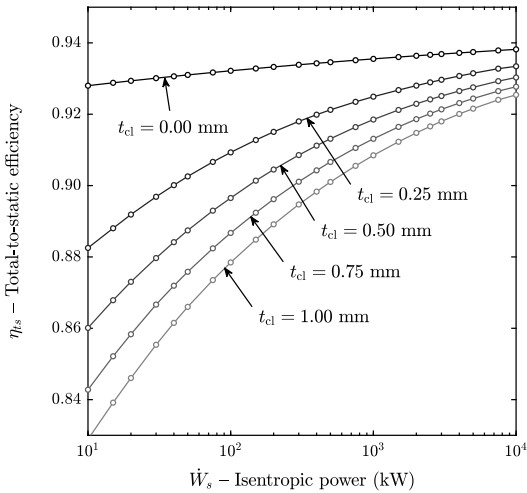
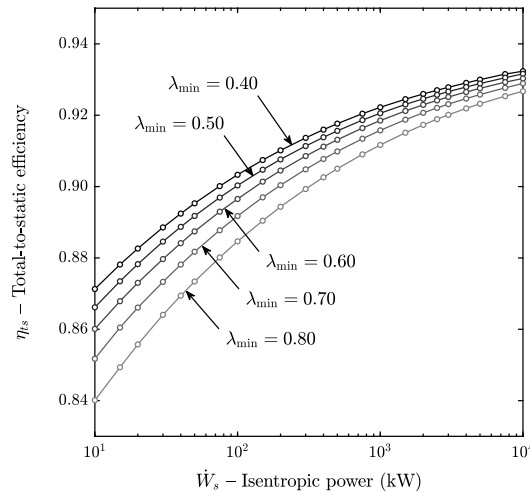

(**a**) Influence of the tip clearance height.          (**b**) Influence of the minimum hub-to-tip ratio constraint.

**Figure 4.** Total-to-static isentropic efficiency as a function of the isentropic power output and (**a**) tip clearance height or (**b**) minimum hub-to-tip ratio constraint.

### 6.2. Influence of Tip Clearance

Figure 4a shows the total-to-static efficiency as a function of the isentropic power when the tip clearance height is varied from 0.00 mm (no clearance) to 1.00 mm (high clearance). It can be observed that the isentropic efficiency decreases when the tip clearance is increased and that the trend is not linear. For instance, increasing the tip clearance from 0.00 mm to 0.25 mm penalizes the efficiency more than from 0.25 to 0.50 mm.

It can also be seen that the efficiency drop due to tip clearance is more marked when the isentropic power is low because the ratio $t_{cl}/H$ is increased both due to blade height reduction and tip clearance increase. In addition, note that total-to-static efficiency increases with the isentropic power output even for the case when the rotor tip clearance is zero due to Reynolds number effects.

### 6.3. Influence of the Hub-to-Tip Ratio

The influence of the lower limit for the hub-to-tip ratio constraint on the turbine performance as a function of the isentropic power is shown in Figure 4b, where the low value $\lambda = 0.40$ is representative of low-pressure steam turbine stages and the high value $\lambda = 0.80$ is representative of gas turbines or high-pressure steam turbine stages. The results of the optimization showed that the constraint for the minimum hub-to-tip ratio is always active at the outlet of the turbine, i.e., the turbine model proposed in this work predicts that the isentropic efficiency will always increase when the allowable lower limit for the hub-to-tip ratio is decreased.

The reason for this is that the blade height is increased when the minimum hub-to-tip ratio decreases and, as a result of this, (1) the tip clearance to blade height ratio $t_{cl}/H$ and the clearance loss coefficient decrease and (2) for a fixed aspect ratio, the blade chord and Reynolds number increase and the profile loss coefficient is reduced. In addition, the channel height of the diffuser increases according to $\hat{b}_{in} = H_{out} / \cos(\phi)$, see Figure 1d. This in turn reduces the friction losses in the diffuser because the channel height appears in the denominator of the friction terms of the diffuser model, Equations (38)–(40). The effect of the hub-to-tip ratio on the friction losses of the diffuser agrees with the results obtained by the authors in a previous work [10], but its impact on the isentropic efficiency is marginal compared with the impact of profile and tip clearance losses.

### 6.4. Influence of the Diffuser Area Ratio

The effect of the diffuser area ratio on the total-to-static isentropic efficiency is shown in Figure 5a,b. The limits of the area ratio were selected to include cases ranging from the absence of diffuser $(AR = 1)$,

to cases where a large fraction of the kinetic energy is recovered ($AR = 5$). This upper limit was selected because, for the case of inviscid, incompressible flow with no inlet swirl, a diffuser with an area ratio of $AR = 5$ would recover 96% of the available dynamic pressure [10].

　　Both Figure 5a,b show that the isentropic total-to-static efficiency increases with the area ratio in an asymptotic way. A small increase of area ratio from the case with no diffuser ($AR = 1$) increases the total-to-static efficiency significantly, whereas, as the area ratio is higher, the improvement of isentropic efficiency becomes less marked because there is less kinetic energy to recover at the diffuser exit. The results of the optimization showed that using an area ratio in the range 2.0–2.5 achieves 70–80% of the maximum efficiency gain. In addition, it was found that the optimum absolute flow angle at the outlet of the last rotor was very close to zero (no swirl) for all cases, regardless of the area ratio of the diffuser.

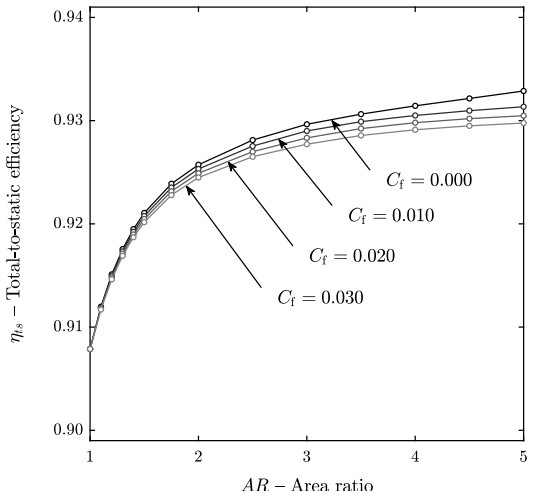
(**a**) Influence of the skin friction coefficient.

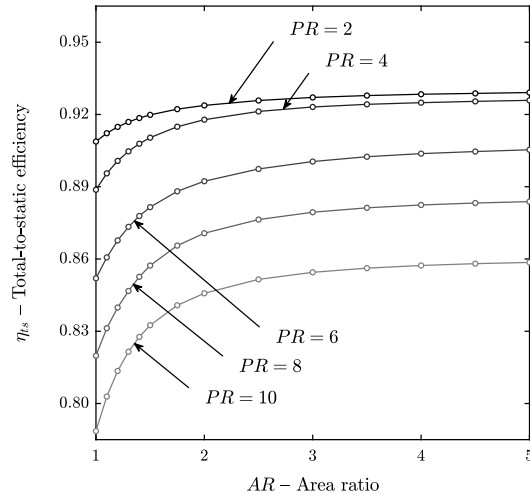
(**b**) Influence of the pressure ratio.

**Figure 5.** Total-to-static isentropic efficiency as a function of the diffuser area ratio and (**a**) skin friction coefficient or (**b**) pressure ratio.

### 6.5. Influence of the Skin Friction Coefficient

　　To the knowledge of the authors, there are no correlations available to predict the skin friction coefficient for annular channels with swirling flow. However, it is possible to estimate a reasonable value for the skin friction coefficient based on experimental data from vaneless diffusers without flow separation. Brown [42] measured the local skin friction coefficient for different vaneless diffusers and obtained values in the range 0.003–0.010. In the absence of better estimates, Johnston and Dean [43] recommend values within the range 0.005–0.010 for the global skin friction coefficient. In a similar way, Dubitsky and Japikse [44] suggest 0.010 as a reasonable estimate for the global skin friction coefficient, but noted that values from 0.005 to 0.020 were required to fit experimental data, depending on the application.

　　In this section, the friction factor was varied from 0.000 (frictionless) to 0.030 (high friction) and the impact on the turbine total-to-static isentropic efficiency is shown in Figure 5a as a function of the area ratio. This range of skin friction factor is representative of well-designed diffusers with attached boundary layers. If the adverse pressure gradient is too high and causes flow separation, the friction losses in the diffuser would increase significantly reducing the pressure recovery and the turbine total-to-static isentropic efficiency [45].

　　It can be observed that increasing the friction factor decreases the total-to-static isentropic efficiency in a linear way (the different curves are equispaced). In addition, the impact of friction factor on the efficiency drop is more notable as the area ratio is high because the length of the channel increases. However, the effect of the friction factor has only a modest impact on the total-to-static efficiency as it causes an efficiency drop of ∼0.3 percentage points for the worst case of Figure 5a.

*6.6. Influence of the Total-to-Static Pressure Ratio*

The effect of the pressure ratio as a function of the diffuser area ratio is shown in Figure 5b. The pressure at the outlet of the turbine was kept constant and the pressure at the inlet was varied to achieve pressure ratios ranging from $PR = 2$ (subsonic flow) to $PR = 10$. It can be observed that increasing the pressure ratio from $PR = 2$ to $PR = 4$ causes a small efficiency drop and that further increasing the pressure ratio to $PR = 6$ causes a much larger efficiency drop. The reason for this is that the Mach number at the outlet of the cascades becomes higher than unity when $PR \approx 4$ and the supersonic correction factor of the Kacker and Okapuu [23] loss system, see Equation (A4), penalizes the total-to-static isentropic efficiency.

In addition, Figure 5b also shows that the total-to-static isentropic efficiency of turbines without diffuser deteriorates rapidly when the pressure ratio is increased. This is because increasing the turbine pressure ratio increases the flow velocities within the turbine and the amount of kinetic energy that is potentially wasted at the outlet. This highlights the importance of using a diffuser when the pressure ratio is high.

*6.7. Influence of the Number of Stages*

Figure 6 shows the total-to-static efficiency of turbines with one, two, and three stages as a function of the pressure ratio. Again, the pressure ratio was achieved varying the pressure at the inlet of the turbine while keeping the outlet pressure constant. However, in this case, the upper limit of the pressure ratio increased to $PR = 14$.

In can be seen that there is a peak of efficiency and that the performance deteriorates rapidly when the pressure ratio increases beyond this point because the flow becomes supersonic and the Mach number correction factor penalizes the profile loss coefficient, see Equation (A4). Moreover, the range of pressure ratios for which the isentropic efficiency is high becomes wider as the number of stages increase because the expansion can be distributed over more cascades and the number of optimization variables increases.

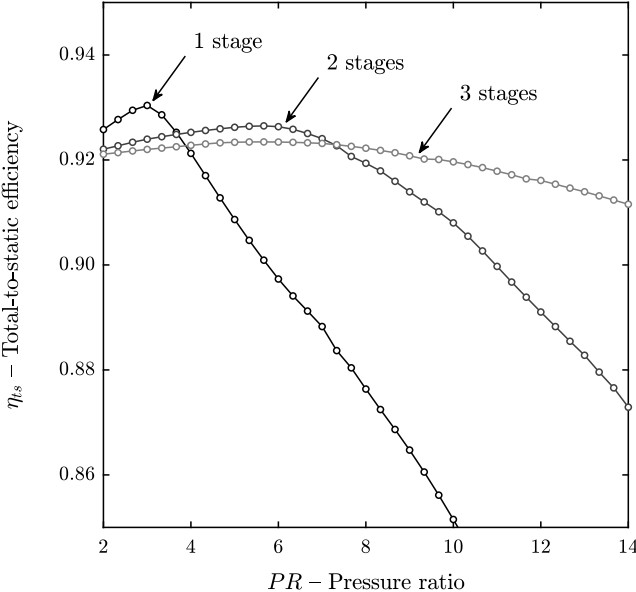

**Figure 6.** Influence of the pressure ratio and number of stages.

*6.8. Influence of the Angular Speed and Diameter*

The results presented in the previous subsections correspond to the optimal values of angular speed and diameter because the specific speed and specific diameter were independent optimization variables with inactive upper and lower bounds. Depending on the application, it might not be

possible to achieve the point of optimal angular speed and diameter because of technical constraints that were not considered in the analysis such as the frequency of the electrical grid, mechanical stress, or space limitations. The objective of this section is to analyze the impact of using a non-optimal angular speed and diameter on the total-to-static efficiency of the turbine. To make the conclusions general, the analysis is presented in terms of dimensionless variables.

Figure 7 shows the contours of maximum total-to-static isentropic efficiency in the $\omega_s$-$d_s$ plane for the 5000 kW reference case of Table 3. In this diagram, often referred as Baljé diagram, the specific speed and specific diameter are regarded as fixed parameters while the rest of the independent optimization variables are free. It can be observed that there exist an optimum specific speed and specific diameter that maximize the total-to-static isentropic efficiency. In addition, there is a narrow region where the efficiency is close to its maximum value and that moving away from this region leads to a rapid decrease in efficiency. Interestingly, the loci of maximum efficiencies are approximately given by the hyperbola of Equation (50).

$$\frac{1}{2}(\omega_s\, d_s) = u/\sqrt{\Delta h_s} = 1 \tag{50}$$

This suggests that the efficiency penalty away from the point of optimal specific speed and specific diameter is small if the dimensionless blade velocity $u/\sqrt{\Delta h_s}$ is close to unity. This simple result can be explained from Euler's turbomachinery equation and the behavior of the solutions that maximize efficiency. On the one hand, close-to-optimal solutions tend to minimize the swirling kinetic energy lost at the exit of the turbine, see Section 6.4. As a consequence, the absolute flow angle and tangential velocity at the rotor exit are close to zero ($\alpha_{\text{out}} \to 0$ and $v_{\theta,\,\text{out}} \to 0$). On the other hand, close-to-optimal solutions also tend to have a relative flow angle at the inlet of the rotor that is close to zero ($\beta_{\text{in}} \to 0$) because the Kacker and Okapuu [23] loss system predicts low profile losses for reaction blades with small relative inlet angles, see Equation (A7). As a result, of this, the absolute tangential velocity at the inlet of the rotor approaches the blade velocity ($v_{\theta\,\text{in}} \to u$).

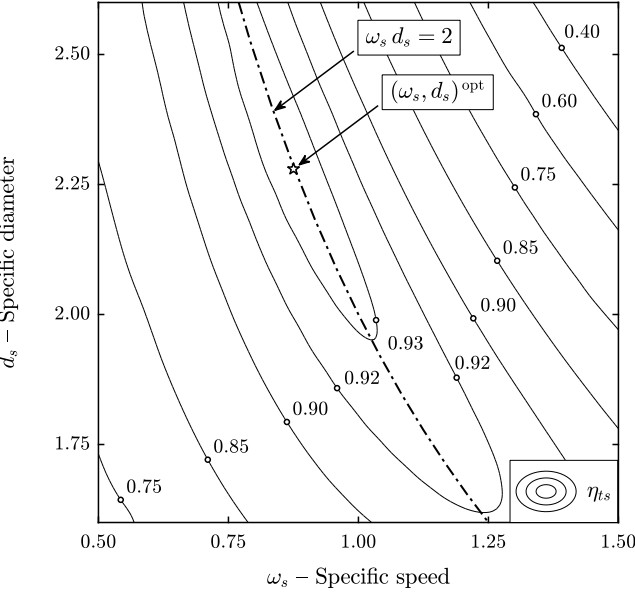

**Figure 7.** Balje diagram of the reference case.

Under these conditions, the actual enthalpy change approaches the isentropic enthalpy change ($\Delta h \to \Delta h_s$) and Euler's turbomachinery equation, Equation (51), is reduced to Equation (52), which corresponds to the hyperbola of maximum efficiencies in the Baljé diagram.

$$\Delta h = [u \, v_\theta]_{\text{in,R}} - [u \, v_\theta]_{\text{out,R}} \tag{51}$$

$$\Delta h_s = u^2 \Rightarrow \frac{u}{\sqrt{\Delta h_s}} = \frac{1}{2}(\omega_s \, d_s) = 1 \tag{52}$$

This analysis was valid for single-stage turbines, but it can be extended to turbines with more than one stage. For the case of a multistage turbine, it was observed that the loci of maximum efficiencies are approximately given by the hyperbola of Equation (53). This relation can also be explained from Euler's turbomachinery when $v_{\theta,\text{out}} \to 0$ and $v_{\theta,\text{in}} \to u$ hold for every stage.

$$\frac{1}{2}(\omega_s \, d_s) = u/\sqrt{\Delta h_s} = \frac{1}{\sqrt{N}} \tag{53}$$

To assess the validity of this result, the optimal blade speed predicted by Equation (53) was compared with the results of numerical optimization for different isentropic power outputs ranging between 10 kW and 10 MW and different pressure ratios ranging between 2 and 14, see Table 5. It can be observed that location of the point of maximum efficiency predicted by Equation (53) agrees well (relative deviation <4%) with the optimization results for axial turbines of 1, 2, and 3 stages regardless of the pressure ratio and the isentropic power output.

**Table 5.** Comparison of the suggested optimal specific blade speed with the optimization results.

| Variable | Sample Points [a] | Number of Stages | Proposed $\frac{1}{2}(\omega_s \, d_s)^{\text{ref}}$ | Optimization [b] $\frac{1}{2}(\omega_s \, d_s)^{\text{mean}}$ | Relative Error [c] |
|---|---|---|---|---|---|
| | 37 | 1 | 1 | 0.978 | 3.22% |
| $2 \leq PR \leq 14$ | 37 | 2 | $1/\sqrt{2}$ | 0.706 | 1.56% |
| | 37 | 3 | $1/\sqrt{3}$ | 0.592 | 0.90% |
| | 23 | 1 | 1 | 1.014 | 2.05% |
| $10\,\text{kW} \leq \dot{W}_s \leq 10\,\text{MW}$ | 23 | 2 | $1/\sqrt{2}$ | 0.725 | 1.42% |
| | 23 | 3 | $1/\sqrt{3}$ | 0.599 | 1.29% |

[a] Number of points ($N$) used to sample the $PR$ and $\dot{W}_s$ intervals. [b] Computed according to $\frac{1}{2N}\sum_{i=1}^{i=N}(\omega_s \, d_s)_i$ where the index $i = 1, 2, \dots, N$ corresponds to each of the sample points. [c] Computed according to $\frac{1}{(\omega_s \, d_s)^{\text{ref}}}\sqrt{\frac{1}{N}\sum_{i=1}^{i=N}\left[(\omega_s \, d_s)_i - (\omega_s \, d_s)^{\text{ref}}\right]^2}$.

## 7. Conclusions

A mean-line model and optimization methodology for axial turbines with any number of stages was proposed. The model was formulated to use arbitrary equations of state and empirical loss models and it accounts for the influence of the diffuser on turbine performance using a one-dimensional flow model proposed by the authors in a previous publication [10]. To the knowledge of the authors, this was the first time that a diffuser model has been coupled with a mean-line model for the optimization of axial turbines. The axial turbine preliminary design was formulated as a constrained optimization problem and was solved using a sequential quadratic programming algorithm. Employing a gradient-based algorithm (instead of a direct search one) allowed to use equality constraints to integrate the cascade, loss, and diffuser sub-models in a simple way.

The model was validated against two test cases from the literature and it was found that the deviation between experimental data and model prediction in terms of mass flow rate and power output was less than 2.5% for both cases and that the deviation in total-to-static efficiency was only 0.27 percentage points for the one-stage case and 0.35 points for the two-stage case. It was also concluded that the close match between measured and predicted efficiencies is probably incidental because the uncertainty of the efficiencies predicted by the loss model is approximately ±1.5 percentage points. In addition, the optimization methodology was applied to a case study from the literature and a

sensitivity analysis was performed to investigate the influence of several design variables on the total-to-static isentropic efficiency, gathering the following conclusions and design guidelines:

- The total-to-static isentropic efficiency increases when the tip clearance height decreases, and this effect is more marked as the isentropic power output of the turbine decreases. This highlights the importance of using small tip clearances in small-scale applications.
- The total-to-static isentropic efficiency increases when the minimum hub-to-tip ratio constraint is reduced (this constraint is always active at the exit of the last rotor). However, reducing the minimum hub-to-tip ratio also increases the centrifugal and gas bending stresses [1]. Therefore, the choice of minimum hub-to-tip ratio must be a trade-off between the fluid-dynamic and the mechanical designs.
- The total-to-static isentropic efficiency increases with the diffuser area ratio in an asymptotic way, regardless of the value of the diffuser skin friction coefficient, and the results of the optimization showed that using an area ratio in the range 2.0–2.5 achieves 70–80% of the maximum efficiency gain. Using a higher diffuser area ratio will increase the kinetic energy recovery and the power output; but it will also increase the turbine footprint, which may be a disadvantage for applications with space limitations.
- The total-to-static isentropic efficiency decreases when the pressure ratio is increased beyond a certain value because the Kacker and Okapuu [23] loss model predicts an increase of the profile loss coefficient when the flow becomes supersonic. This effect becomes less marked as the number of stages increases because the expansion can be distributed over more cascades. In addition, the total-to-static efficiency of turbines without diffuser deteriorates rapidly when the pressure ratio is increased, highlighting the importance of using a diffuser when the pressure ratio is high.
- The results of the optimization showed that the maximum total-to-static isentropic efficiency is attained when the absolute flow angle at the exit of the last stage is close to zero (no exit swirl), regardless of the area ratio of the diffuser. This agrees with the conclusions drawn in a previous work from the authors where the flow within the diffuser was examined in more detail [10].
- It was found that the efficiency penalty away from the point of optimal angular speed and diameter, peak of the Baljé diagram, is small if the combination of specific speed and diameter is close to the hyperbola given by $\omega_s \, d_s = 2/\sqrt{N}$. This guideline can be used to select a suitable combination of angular speed and diameter when one of these variables is imposed by technical constraints such as the frequency of the electrical grid, mechanical stress, or space limitations.

**Supplementary Materials:** The source code with the implementation of the mean-line model and optimization methodology described in this paper is available in an online repository (doi:10.5281/zenodo.2635586).

**Author Contributions:** Conceptualization, R.A. and L.O.N.; methodology, R.A.; software, R.A.; validation, R.A.; resources, L.O.N.; writing–original draft preparation, R.A.; writing–review and editing, R.A., L.O.N.

**Funding:** The authors gratefully acknowledge the financial support from the Research Council of Norway (EnergiX grant no. 255016) for the COPRO project, and the user partners Equinor, Hydro, Alcoa, GE Power Norway and FrioNordica.

**Acknowledgments:** The authors gratefully acknowledge the support from Roberto Pili for his help in the validation of the axial turbine model.

**Conflicts of Interest:** The authors of the article, Roberto Agromayor and Lars O. Nord, declare that they have no conflict of interest.

## Nomenclature
**Latin symbols**

| | | |
|---|---|---|
| $a$ | Speed of sound | m/s |
| $A$ | Flow area | m$^2$ |
| $AR$ | Diffuser area ratio | – |
| $b$ | Blade axial chord | m |
| $\hat{b}$ | Diffuser channel height | m |

| | | |
|---|---|---|
| $c$ | Blade chord | m |
| $C_f$ | Diffuser skin friction coefficient | – |
| $d$ | Turbine mean diameter | m |
| $d_s$ | Turbine specific diameter | – |
| $h$ | Static specific enthalpy | J/kg |
| $h_0$ | Stagnation specific enthalpy | J/kg |
| $\Delta h_s$ | Total-to-static isentropic specific enthalpy change | J/kg |
| $H$ | Blade height | m |
| $\dot{m}$ | Mass flow rate | kg/s |
| $N$ | Number of turbine stages | – |
| $o$ | Blade opening | m |
| $p$ | Static pressure | Pa |
| $p_0$ | Stagnation pressure | Pa |
| $PR$ | Total-to-static pressure ratio | – |
| $\dot{q}_w$ | Heat flux at the diffuser wall | $W/m^2$ |
| $r$ | Turbine mean radius | m |
| $\hat{r}$ | Diffuser mean radius | m |
| $r_h$ | Radius at the hub of the blades | m |
| $r_t$ | Radius at the tip of the blades | m |
| $s$ | Blade pitch or specific entropy | m or $J/kg\,K$ |
| $s_c$ | Cascade spacing | m |
| $T$ | Static temperature | K |
| $T_0$ | Stagnation temperature | K |
| $t_{cl}$ | Tip clearance height | m |
| $t_{max}$ | Maximum blade thickness | m |
| $t_{te}$ | Trailing edge thickness | m |
| $u$ | Blade velocity | m/s |
| $v$ | Absolute flow velocity | m/s |
| $v_0$ | Isentropic velocity (also known as spouting velocity) | m/s |
| $w$ | Relative flow velocity | m/s |
| $\dot{W}$ | Actual power output | W |
| $\dot{W}_s$ | Isentropic power output | W |
| $Y$ | Stagnation pressure loss coefficient | – |

**Greek symbols**

| | | |
|---|---|---|
| $\alpha$ | Absolute flow angle | ° |
| $\beta$ | Relative flow angle | ° |
| $\delta$ | Deviation angle or diffuser semi-divergence angle | ° |
| $\delta_{fl}$ | Blade flaring angle | ° |
| $\eta_{ts}$ | Total-to-static isentropic efficiency | – |
| $\Delta\theta$ | Camber angle | ° |
| $\theta$ | Metal angle | ° |
| $i$ | Incidence angle | ° |
| $\lambda$ | Hub-to-tip radii ratio | – |
| $\mu$ | Dynamic viscosity | Pa s |
| $\zeta$ | Stagger angle (also known as setting angle) | ° |
| $\rho$ | Density | $kg/m^3$ |
| $\tau_w$ | Shear stress at the diffuser wall | Pa |
| $\phi$ | Diffuser mean wall cant angle or kinetic energy loss coefficient ($\phi^2 + \Delta\phi^2 = 1$) | ° or – |
| $\omega$ | Angular speed | rad/s |
| $\omega_s$ | Specific speed | – |

**Abbreviations**

| | |
|---|---|
| CFD | Computational Fluid Dynamics |
| ODE | Ordinary Differential Equation |

**Subscripts**

| | |
|---|---|
| 0 | Stagnation state |
| 1 | Inlet of the turbine |
| 2 | Outlet of the turbine |
| in | Inlet of the cascade |
| out | Outlet of the cascade |
| error | Violation of an equality constraint |
| *m* | Meridional direction |
| *r* | Radial direction |
| ref | Turbine exit state assuming a reference isentropic efficiency |
| rel | Relative to the rotating frame of reference |
| *s* | Isentropic expansion |
| *x* | Axial direction |
| *θ* | Tangential direction |

**Appendix A. Kacker–Okapuu Loss Model**

This appendix describes the loss model proposed by Kacker and Okapuu [23] to compute aerodynamic losses in axial turbines. This model is a refinement of the correlations proposed by Ainley and Mathieson [26,27] and by Dunham and Came [28]. The general form of the Kacker-Okapuu loss system given by Equation (A1).

$$Y = f_{\mathrm{Re}}\, f_{\mathrm{Ma}}\, Y_{\mathrm{p}} + Y_{\mathrm{s}} + Y_{\mathrm{cl}} + Y_{\mathrm{te}} \tag{A1}$$

The expressions used to compute each term of this equation as a function of the cascade geometry and the thermodynamic and kinematic variables of the flow are presented in the next subsections. Some of the signs from the original correlations were modified to comply with the angle convention used in this work. These modifications are explicitly mentioned in the text.

*Appendix A.1. Reynolds Number Correction Factor*

The term $f_{\mathrm{Re}}$ accounts for the effects of the Reynolds number and it is computed according to Equation (A2).

$$f_{\mathrm{Re}} = \begin{cases} \left(\frac{\mathrm{Re}}{2\cdot 10^5}\right)^{-0.40} & \text{for} \quad \mathrm{Re} < 2\cdot 10^5 \\ 1 & \text{for} \quad 2\cdot 10^5 < \mathrm{Re} < 1\cdot 10^6 \\ \left(\frac{\mathrm{Re}}{1\cdot 10^6}\right)^{-0.20} & \text{for} \quad \mathrm{Re} > 1\cdot 10^6 \end{cases} \tag{A2}$$

The Reynolds number is given by Equation (A3) and it is defined in terms of the chord length and the density, viscosity, and relative velocity at the outlet of the cascade.

$$\mathrm{Re} = \frac{\rho_{\mathrm{out}}\, w_{\mathrm{out}}\, c}{\mu_{\mathrm{out}}} \tag{A3}$$

*Appendix A.2. Mach Number Correction Factor*

The term $f_{\mathrm{Ma}}$ accounts for losses associated with supersonic flows at the trailing edge of the blades and it is computed according to Equation (A4).

$$f_{\mathrm{Ma}} = \begin{cases} 1 & \text{for} \quad \mathrm{Ma}_{\mathrm{out}}^{\mathrm{rel}} \leq 1 \\ 1 + 60\cdot (\mathrm{Ma}_{\mathrm{out}}^{\mathrm{rel}} - 1)^2 & \text{for} \quad \mathrm{Ma}_{\mathrm{out}}^{\mathrm{rel}} > 1 \end{cases} \tag{A4}$$

The Mach number is given by Equation (A5) and it is defined by the relative velocity and the speed of sound at the outlet of the cascade.

$$\mathrm{Ma}_{\mathrm{out}}^{\mathrm{rel}} = w_{\mathrm{out}}/a_{\mathrm{out}} \tag{A5}$$

*Appendix A.3. Profile Loss Coefficient*

The profile loss coefficient $Y_p$ is computed according to Equation (A6).

$$Y_p = 0.914 \cdot \left( \frac{2}{3} \cdot Y_p' \cdot K_p + Y_{shock} \right) \tag{A6}$$

The term $Y_p'$ is given by Equation (A7), where the terms, $Y_{p,\,reaction}$ and $Y_{p,\,impulse}$ are be obtained from the graphical data reproduced in Figures A1 and A2. The subscript *reaction* refers to blades with zero inlet metal angle (axial entry) and the subscript *impulse* refers to blades that have an inlet metal angle with the same magnitude but opposite sign as the exit relative flow angle. The second term of the right-hand side of Equation (A7) is a correction factor that accounts for the effect of the maximum blade thickness. The sign of $\beta_{out}$ in Equation (A7) was changed with respect to the original work of Kacker–Okappu to comply with the angle convention used in this paper.

$$Y_p' = \left[ Y_{p,\,reaction} - \left( \frac{\theta_{in}}{\beta_{out}} \right) \left| \frac{\theta_{in}}{\beta_{out}} \right| \cdot (Y_{p,\,impulse} - Y_{p,\,reaction}) \right] \cdot \left( \frac{t_{max}/c}{0.20} \right)^{-\frac{\theta_{in}}{\beta_{out}}} \tag{A7}$$

The factor $K_p$ from Equation (A6) accounts for compressible flow effects when the Mach number within the cascade is subsonic and approaches unity. These effects tend to accelerate the flow, make the boundary layers thinner, and decrease the profile losses. $K_p$ is a function on the inlet and outlet relative Mach numbers and it is computed from Equations (A8)–(A10).

$$K_p = 1 - K_2 \cdot (1 - K_1) \tag{A8}$$

$$K_1 = \begin{cases} 1 & \text{for} \quad \mathrm{Ma}_{out}^{rel} < 0.20 \\ 1 - 1.25 \cdot (\mathrm{Ma}_{out}^{rel} - 0.20) & \text{for} \quad 0.20 < \mathrm{Ma}_{out}^{rel} < 1.00 \\ 0 & \text{for} \quad \mathrm{Ma}_{out}^{rel} > 1.00 \end{cases} \tag{A9}$$

$$K_2 = \left( \frac{\mathrm{Ma}_{in}^{rel}}{\mathrm{Ma}_{out}^{rel}} \right)^2 \tag{A10}$$

The term $Y_{shock}$ from Equation (A6) accounts for the relatively weak shock waves that may occur at the leading edge of the cascade due to the acceleration of the flow. After some algebra, the equations proposed in the Kacker–Okapuu method can be summarized as Equation (A11), where $f_{hub}$ is given graphically in Figure A3 and it is a function of the hub-to-tip ratio only. Please note that the nomenclature used in Kacker and Okapuu [23] is different than then one used in this work, in particular $q \equiv p_{0rel} - p$ and $\Delta P \equiv p_{0rel,in} - p_{0rel,out}$.

$$Y_{shock} = 0.75 \cdot \left( f_{hub} \cdot \mathrm{Ma}_{in}^{rel} - 0.40 \right)^{1.75} \cdot \left( \frac{r_{hub}}{r_{tip}} \right)_{in} \cdot \left( \frac{p_{0rel,in} - p_{in}}{p_{0rel,out} - p_{out}} \right) \tag{A11}$$

*Appendix A.4. Secondary Loss Coefficient*

The secondary loss coefficient $Y_s$ is computed according to Equation (A12).

$$Y_s = 1.2 \cdot K_s \cdot \left[ 0.0334 \cdot f_{AR} \cdot Z \cdot \left( \frac{\cos(\beta_{out})}{\cos(\theta_{in})} \right) \right] \tag{A12}$$

The factor 1.2 is included to correct the secondary loss for blades with zero trailing edge thickness. Trailing edge losses are accounted independently.

The factor $K_s$ accounts for compressible flow effects when the Mach number within the cascade is subsonic and approaches unity. These effects tend to accelerate the flow, make the end wall boundary

layers thinner, and decrease the secondary losses. $K_s$ is computed from Equation (A13), where $K_p$ is given by Equation (A8) and $K_3$ is given by Equation (A14). $K_3$ is a function of the axial blade aspect ratio $H/b$ only.

$$K_s = 1 - K_3 \cdot \left(1 - K_p\right) \tag{A13}$$

$$K_3 = \left(\frac{1}{H/b}\right)^2 \tag{A14}$$

$f_{AR}$ accounts for the blade aspect ratio $H/c$ and it is given by Equation (A15).

$$f_{AR} = \begin{cases} \frac{1 - 0.25 \cdot \sqrt{2 - H/c}}{H/c} & \text{for} \quad H/c < 2 \\ \frac{1}{H/c} & \text{for} \quad H/c > 2 \end{cases} \tag{A15}$$

The Ainley-Mathieson loading parameter $Z$ is given by Equations (A16)–(A18), where the sign of $\beta_{out}$ was changed with respect to the original work of Kacker and Okapuu [23] to comply with the angle convention used in this paper.

$$Z = \left(\frac{C_L}{s/c}\right)^2 \frac{\cos(\beta_{out})^2}{\cos(\beta_m)^3} \tag{A16}$$

$$\left(\frac{C_L}{s/c}\right) = 2\cos(\beta_m)\left[\tan(\beta_{in}) - \tan(\beta_{out})\right] \tag{A17}$$

$$\tan(\beta_m) = \frac{1}{2}\left[\tan(\beta_{in}) + \tan(\beta_{out})\right] \tag{A18}$$

*Appendix A.5. Tip Clearance Loss Coefficient*

The clearance loss coefficient $Y_{cl}$ is computed according to Equation (A19), where the influence of the number of seals is neglected.

$$Y_{cl} = B \cdot Z \cdot \left(\frac{c}{H}\right) \cdot \left(\frac{t_{cl}}{H}\right)^{0.78} \tag{A19}$$

In this equation, $Z$ is given by Equations (A16)–(A18). The Kacker-Okapuu loss system proposes $B = 0.37$ for rotor blades with shrouded tips, and $B = 0.00$ for stator blades. In addition, Kacker and Okapuu warn that using $B = 0.47$, as suggested by Dunham and Came [28], over-predicts the loss for rotor blades with plain tips.

*Appendix A.6. Trailing Edge Loss Coefficient*

The trailing edge loss coefficient $Y_{te}$ is computed according to Equation (A20).

$$Y_{te} \approx \zeta = \frac{1}{\phi^2} - 1 = \frac{1}{1 - \Delta\phi^2} - 1 \tag{A20}$$

where the pressure loss coefficient $Y$ was approximated by the enthalpy loss coefficient $\zeta$ and then related to the kinetic energy loss coefficients $\phi^2$ and $\Delta\phi^2$. See the work by Dahlquist [36] for details about the definitions of the different loss coefficients and the relations among them. The parameter $\Delta\phi^2$ is computed by interpolation of impulse and reaction blades according to Equation (A21). The sign of $\beta_{out}$ in Equation (A21) was changed with respect to the original work of Kacker–Okappu to comply with the angle convention used in this paper.

$$\Delta\phi^2 = \Delta\phi^2_{reaction} - \left(\frac{\theta_{in}}{\beta_{out}}\right)\left|\frac{\theta_{in}}{\beta_{out}}\right| \cdot \left(\Delta\phi^2_{impulse} - \Delta\phi^2_{reaction}\right) \tag{A21}$$

$\Delta\phi^2_{\text{reaction}}$ and $\Delta\phi^2_{\text{impulse}}$ are the kinetic energy loss coefficients of reaction and impulse blades and they are a function of the trailing edge thickness to opening ratio $t_{\text{te}}/o$ only. The functional relation was given graphically, and it is reproduced in Figure A4.

*Appendix A.7. Final Remarks*

The Kacker–Okapuu loss model was developed to estimate the performance of competent turbine designs and its predictions will not be accurate if the input parameters are outside the range of the experimental data used to develop the correlations. This situation is often encountered before the optimization algorithm converges since, in general, it is not possible to satisfy constraints for each iterate of a nonlinear programming problem. For this reason, some of the variables used within the Kacker–Okapuu loss model were bounded to avoid numerical problems that might prevent the convergence to a feasible solution. For instance, some variables were forced to be non-negative because the correlations were not developed to cover such cases. These modifications do not affect the final results of the optimization and they are not reported in this paper although they are documented in detail within the code.

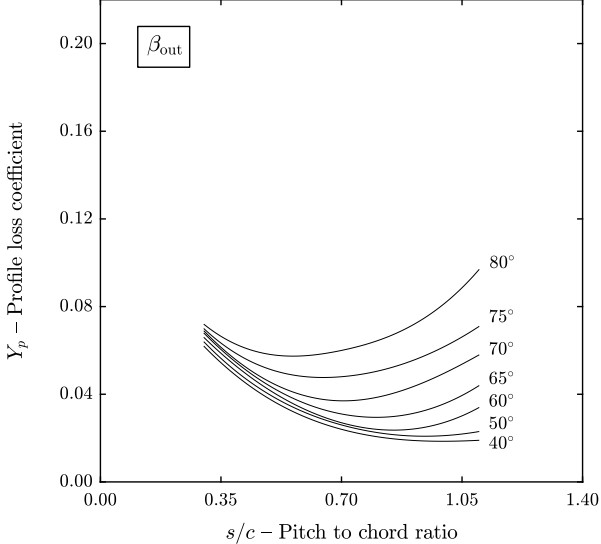

**Figure A1.** Profile loss of reaction blades (axial entry blades).

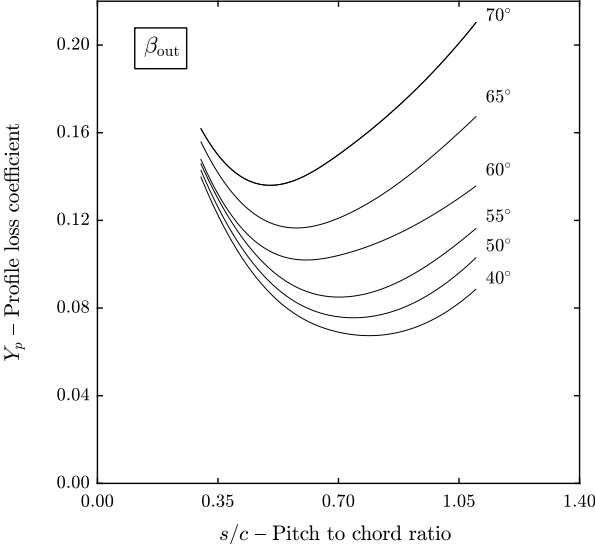

**Figure A2.** Profile loss of impulse blades.

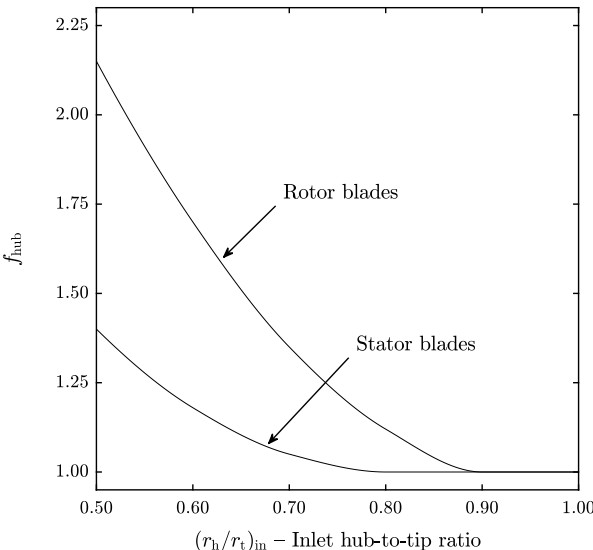

**Figure A3.** Ratio of Mach number at the hub to Mach number at the mean radius.

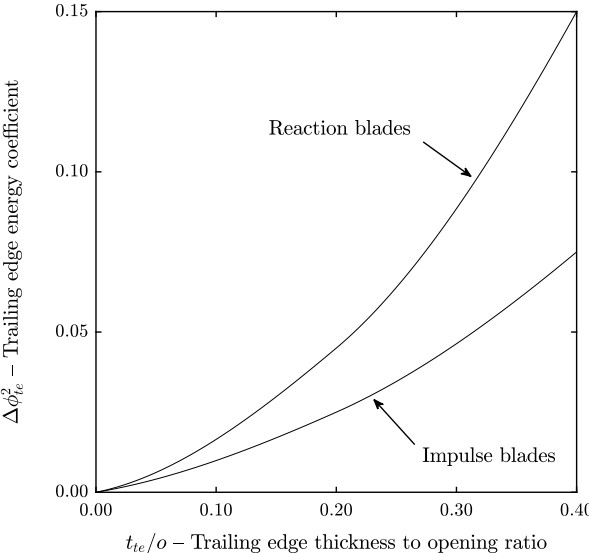

**Figure A4.** Trailing edge energy loss coefficient for impulse and reaction blades.

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
