# Peer review of "Preliminary Design and Optimization of Axial Turbines Accounting for Diffuser Performance"

_ijtpp, doi:10.3390/ijtpp4030032_

Round 1

Reviewer 1 Report

Overall comments

The paper deals with a mean line approach, coupled to the Okapuu’s loss correlations model. Its validation is quite weak, only 2 subsonic test cases presented that do not cover the exploration range of the sensibility analysis depicted in section 6. If available, addition validation test cases would be welcomed. A lot of details are given to explain how the mean line model works. However, part of them (for instance, the definition of the velocity triangle in section 2.4.1) could be simplified in favor of a better description of the sensibility analysis assumptions and results:

About the assumptions: What about the degree of reaction of the turbine, is it an input parameter or a result of the optimization process? In others words, which assumption is used to specify the static pressure ratio between the stator and the rotor?

About the results: For each studied design parameters (isentropic work, tip clearance, hub-to-tip ratio…), I recommend to add new figures illustrating the impact of the geometrical parameters (blade height H, blade chord …) on the efficiency variations. Regarding these geometrical parameters (chord, height…), are they related to the stator? To the rotor? Or to both? 

Detailed comments

 Section “2 - axial turbine geometry”

From figure 3, it seems that the mean line model is restricted to a constant mean radius through the axial turbine, and no explicit description regarding this point seems have been given in the text. Please, be more specific about this point. Although the Okapuu’s correlations model is well defined in the appendix A, a short description of the scope of these correlations would be welcome in the main paper.

Section “2.4 – Cascade model”

The axial spacing parameter (Sc) is defined in figure 2, but no assumption is provided regarding the flow behavior between 2 blades rows, especially regarding losses generation. Please, comment this point.

 Section « 2.4.2. Thermodynamic properties »

Line 159 “For purely axial turbines, the conservation of rothalpy is reduced to the conservation of relative stagnation enthalpy”: I suggest to be more specific about the meaning of “purely axial turbines” à Does it mean “a constant axial mean radius”?

Section “2.5. Loss model »

I’m not sure to fully understand the following statementThis definition is valid both for stator and rotor cascades since the relative stagnation pressure for stator cascades is equal to the absolute stagnation pressure”. Indeed, Equation 37 is valid for stator and also for rotor with constant mean radius. For rotor with a radius variation from the inlet to the outlet, this equation should be modified to account of the “free work” coming from the radius variation. “In general, the loss coefficient computed using its definition, Eq. (37), and the loss coefficient computed using the loss model, Appendix A, will not match”. The meaning of this statement is not cleat at all, please be more specific.

Section 6 “sensibility analysis”

It is not clear at this stage, if the sensibility analysis was performed with the fixed parameters defined in table 3 and if the sensibility analysis is the result of an optimization process? This point could be more explicitly mentioned.

Section 6.3 “influence of the hub-to-tip ratio”

Please, explain what do you mean by “the Influence of the minimum hub-to-tip ratio constraint”? What is the meaning of the word “minimum”? Does this “hub-to-tip” ratio refer to the turbine outlet?

Section 6.6 “Influence of the total-to-static pressure ratio

Line 419: the same word is used twice “penalizes the the total-to-static isentropic efficiency”

 Section 6.8 “Influence of the angular speed and diameter”

The demonstration proposed to explain the relation 52 using the Euler’s equation isn’t clear. Please, try to improve this part (from lines 445 to 449) . Please, provide additional information that illustrates the fact that equation 52 is related to the square root of the number of turbine stages. Please, be more specific about the meaning of the “sample points” column (table 5)

 Section “Nomenclature”

The symbol “e” for “internal energy’ (line 525) does not seem to be used in the paper The symbol ‘Φ” defined as the “diffuser mean wall cant angle” (line 569), has also get the meaning of the “kinetic energy loss coefficients” (line 691 – Equation A.26).

Section “Appendix A. Kacker-Okapuu loss model”

The same word “the” was used twice (line 676) à “This appendix describes the the loss model proposed by” Please, be more specific about the application domain of the profile loss coefficient Yp (line 683 – Equation A.9) as referred in the paper [23] à “Subsonic domain” Equation A-10: A short definitions of “impulse” and reaction” would be welcomed to avoid misunderstanding of the loss correlations application. Starting from the referenced paper [23], it is not clear how the formula defined by the equation A-16 has been obtained, in particular the term fhub inside the brackets. Please, provide additional information helping to convert the original formulation from [23] to the proposed one (Equation A-16). In the reference paper [23], the coefficient K3 is referred to the blade axial chord bx, while in the present paper the blade chord b is used à Please, check the formulation of equation A-19 against those defined in reference [23].

 Section “Appendix A.5 - Tip clearance loss coefficient” (line 687):

The value of the coefficient B is defined first with the value 0.37 and later on, gets the value 47 à Please check these values. The tip clearances loss coefficient proposed in the paper (Equation A-25) differs from the one suggested by Okapuu [23], in particular regarding the term related to the tip gap tcl which is associated with the number of seals (see paper [23] – page 116 – equation 20]. Please, explain such a deviation from the original Okapuu’s tip clearance correlation.

Author Response

See the attached document.

Reviewer 2 Report

In the present paper, a mean-line methodology for the preliminary design of axial turbines is presented. The problem is well introduced, the methodology is described correctly and the topic is of interest. Form a formal point of view I have no comments.

I have reviewed also Part 1, and with respect to that paper I have more concerns. For sure Part 1 is more interesting and useful. I understand that in the present case the optimization is performed for a real gas and that the authors validated the model for a realistic case (section 3), but in this case the risk is to think about a work done in the eighties. In our research group we did a similar work during a master thesis two years ago and the level of complexity of this work is not much higher.

Anyway, I recognize the effort to develop a fast and reliable tool (which is also free for download, but I did not test it for lack of time) for the arbitrary optimization of axial turbines with real gas. I also appreciate the full presentation of the Kacker-Okapuu loss model: I think that correlations should not be forgot too early, substituting them with compulsive CFD campaigns. I think that this paper will be useful for anybody who want to develop his own code and for students who are searching for such kind of model for their activities.

I can only suggest to provide more information about the optimization procedure, the necessary time for an optimization procedure and the generated (discarded) geometries. Those information are usually present a paper that deals with optimization.

Apart from that, I have no more comments and I support the publication of the paper after that small (not mandatory) modification.

Author Response

We appreciate the positive comments from the reviewer and we are happy that the reviewer supports the publication.